# Microsatellite Instability: From the Implementation of the Detection to a Prognostic and Predictive Role in Cancers

**DOI:** 10.3390/ijms23158726

**Published:** 2022-08-05

**Authors:** Martina Amato, Renato Franco, Gaetano Facchini, Raffaele Addeo, Fortunato Ciardiello, Massimiliano Berretta, Giulia Vita, Alessandro Sgambato, Sandro Pignata, Michele Caraglia, Marina Accardo, Federica Zito Marino

**Affiliations:** 1Pathology Unit, Department of Mental and Physical Health and Preventive Medicine, University of Campania “L. Vanvitelli”, 80138 Naples, Italy; 2Medical Oncology Unit, SM delle Grazie Hospital, 80078 Pozzuoli, Italy; 3Medical Oncology Unit, San Giovanni di Dio Hospital, 80027 Frattamaggiore, Italy; 4Medical Oncology, Department of Precision Medicine, Università degli Studi della Campania “Luigi Vanvitelli”, 80131 Naples, Italy; 5Department of Clinical and Experimental Medicine, University of Messina, 98121 Messina, Italy; 6Anatomical Pathology Department, IRCCS CROB, 85028 Rionero in Vulture, Italy; 7Scientific Direction, Centro di Riferimento Oncologico della Basilicata (IRCCS-CROB), 85028 Rionero in Vulture, Italy; 8Division of Medical Oncology, Department of Uro-Gynaecological Oncology, Istituto Nazionale per lo Studio e la Cura dei Tumori “Fondazione G. Pascale”, IRCCS, 80131 Naples, Italy; 9Department of Precision Medicine, University of Campania “L. Vanvitelli”, 80138 Naples, Italy

**Keywords:** microsatellite instability, mismatch repair, immunotherapy, immunohistochemistry, PCR, NGS

## Abstract

Microsatellite instability (MSI) has been identified in several tumors arising from either germline or somatic aberration. The presence of MSI in cancer predicts the sensitivity to immune checkpoint inhibitors (ICIs), particularly PD1/PD-L1 inhibitors. To date, the predictive role of MSI is currently used in the selection of colorectal cancer patients for immunotherapy; moreover, the expansion of clinical trials into other cancer types may elucidate the predictive value of MSI for non-colorectal tumors. In clinical practice, several assays are used for MSI testing, including immunohistochemistry (IHC), polymerase chain reaction (PCR) and next-generation sequencing (NGS). In this review, we provide an overview of MSI in various cancer types, highlighting its potential predictive/prognostic role and the clinical trials performed. Finally, we focus on the comparison data between the different assays used to detect MSI in clinical practice.

## 1. Introduction

Microsatellites (MSs) are repeated sequences of 1–6 nucleotides in the human genome. MSs are highly polymorphic; in particular, their distribution and their dimensions are variable [1,2]. MSs are widely used both as markers in studies of genetic disease and as predictive biomarkers in various cancer types [3,4,5]. MSs are often subject to the incorporation or deletion of repeated units made by DNA polymerase during DNA replication; the mismatch repair system (MMR) is generally able to correct these errors [6]. The MMR is a pivotal system for maintaining genome integrity, consisting of several proteins, such as MLH1, MLH3, MSH2, MSH6, MSH3, PMS2, PMS1 and Exo1. The DNA errors are detected by MSH2/MSH6 and MSH2/MSH3 heterodimers; subsequently, the MLH1/PMS2 complex nicks DNA at sites of mismatch in order to begin strand repair [7]. MLH1 and MSH2 are the primary partners of their heterodimer, while PMS2 and MSH6 are the respective secondary partners. Consequently, the loss of the primary partners results in the loss of the entire heterodimer, but not the opposite. The alteration of the MMR proteins leads to a biological status known as microsatellite instability (MSI) [8]. The generation of MSI can be caused by the point mutations occurring in MMR genes, the DNA polymerase slippage during the replication process and the insertion/deletion of one or more bases in the microsatellite regions [1]. Furthermore, the MSI status could be due to other aberrations, including the hypermethylation of the MLH1 promoter [9], the epigenetic inactivation of MSH2 or MLH1 [10,11], the downregulation of MMR genes by microRNAs [12] or the slipped strand mating error (SSM) [13]. The MSI phenotype could be related to Lynch syndrome (LS), a genetic disease characterized by inherited germline mutations in MMR genes that is mainly shown in colorectal cancer and endometrial cancer; however, it is also shown in ovarian cancer, gastric cancer, the hepatobiliary tract, upper urinary tract, pancreas, brain and skin [14,15,16,17]. The presence of MSs in the coding regions is prone to generate genetic instability, leading to carcinogenesis [18]. Oda et al. proposed MSI classification into type A, with a variation of <6 bp, and type B, with a variation of >8 bp. Thibodeau et al. classified MSI based on the increase or decrease in fragment size into type I and II, respectively [19,20]. MSI was identified for the first time in colorectal cancer (CRC) [21]; subsequently, it was reported in endometrial cancer, ovarian cancer and gastric cancer (Figure 1). In recent years, MSI has also been identified with less frequency in other tumors, such as non-small cell lung cancer, breast cancer, brain cancer, prostate cancer, pancreatic cancer, bladder cancer and melanoma [22,23,24,25,26]. The immunocheckpoint inhibitors (ICI) have been shown to be effective in several tumor types carrying MSI-H compared to chemotherapy [27,28,29]. The rationale behind the sensitivity to ICI of MSI-H tumors is probably due to the accumulation of mutations that determines new mutated peptides, leading to the activation of the immune system [30]. In this context, MSI tumors have a high level of T-helper 1/cytotoxic lymphocytes and immune checkpoint molecules such as CTLA4, PD-1, PD-L1, LAG-3 and IDO, compared to MSS tumors [31,32]. The tumors carrying MSI showed a much better response to ICI than MSS cancers [33]. In recent years, the immunotherapy has achieved excellent results in several cancer types with the MSI phenotype [34]. In 2017, the US Food and Drug Administration (FDA) approved pembrolizumab, a PD-1 inhibitor, for the treatment of MSI advanced solid tumors regardless of the cancer type and the histology. MSI is the first cancer type-agnostic biomarker approved to establish the eligibility of patients for treatment with ICI rather than the site of the tumor [27].

## 2. Methods of MSI Detection

The MSI implication in cancer leads researchers to develop several techniques for its detection. The MSI phenotype could be defined both by a molecular approach evaluating the microsatellite sequences and by IHC evaluating the expression of MMR proteins (MMRP). The immunohistochemistry (IHC) analyzes the expression of a limited number of MMR proteins, including MLH1, PMS2, MSH2 and MSH6. MLH1 and MSH2 are the primary partners, and their degradation leads to the loss of their respective partners [35]. The proficient MMR (pMMR) cases show positive staining of the nuclei of all four MMRP; the deficient MMR (dMMR) cases show the loss of one of the two MLH1/PMS2 or MSH2/MSH6 heterodimers; the patchy MMR cases show focal IHC staining or cytoplasmic staining of one or more MMRP [36,37]. Some studies suggest that IHC could be exclusively used as a screening method for MSI detection since it is fast, easy to perform, inexpensive and widespread to most pathological laboratories [38,39,40,41]. Previous findings showed that both technical and interpretative pitfalls affect the MMR IHC, leading to false IHC results [42]. MMR IHC may be affected by pre-analytical problems, particularly the artifacts due to inadequate fixation. The adequate IHC staining interpretation requires the evaluation of a positive internal control, such as a normal stroma, and external control tissue, such as intratumoral lymphocytes and fibroblasts [43,44]. Moreover, a heterogeneous MMR IHC staining pattern could be caused by non-truncating mutations (missense mutations or frame insert/delete mutations) that could lead to full-length MMRP not functioning [41,45].

MSI molecular testing is generally performed by polymerase chain reaction (PCR). MSI-PCR is currently performed through two commercial panels, the Bethesda and the Pentaplex. The Bethesda panel includes five predefined genomic regions: two single-nucleotide (BAT25, BAT-26) and three dinucleotides (D2S123, D5S346 and D17S250). The Pentaplex panel includes five single-nucleotide loci, including BAT-25, BAT-26, NR-21, NR-24 and NR-27 [46,47]. The microsatellite status can be classified according to the number of instability markers, as follows: (i) high microsatellite instability (MSI-H) when there are more than 2 unstable loci; (ii) low microsatellite instability (MSI-L) when there is only one unstable locus; and (iii) microsatellite stability (MSS) when all the microsatellites are stable [48]. MSI-L and MSS tumors have similar molecular and clinical features; thus, patients carrying MSI-L are classified as MSS and they are not sensitive to ICI compared to patients carrying MSI-H [1]. MSI-PCR is a highly sensitive technique that is able to identify the instability due to the non-truncating mutations generally associated with a false positive IHC staining [41].

To date, both IHC and PCR are used in the clinical practice of MSI testing to define patients who can benefit from ICI treatment. However, several data in the literature showed discordant results between IHC and PCR in MSI detection. Several studies demonstrated that MMR IHC results frequently have not been confirmed by molecular analysis. For example, some cases of dMMR-IHC are results of MSS by PCR, resulting in clinical implications in the therapeutic choice. Likewise, cases with pMMMR-IHC were unexpectedly MSI by PCR [41,49,50,51,52]. These controversial results are generally attributable to the technical limits related to MMR IHC or a misinterpretation of the staining [41].

Recently, the next-generation sequencing (NGS) is another method used for MSI detection [2,53]. Several NGS-based methods have been described to assess MSI, including MSIseq somatic mutation analysis [54], mSINGS [53], MSISensor [55], Microsatellite Analysis for Normal Tumor Instability (MANTIS) [56], ColoSeq [57] and MSI-ColonCore [58]. The main advantage of an NGS-based approach is the simultaneous screening of microsatellite loci and other mutations including TMB [59].

In conclusion, tumors with the loss or equivocal staining of at least one of the MMRP by IHC require an additional molecular test in order to avoid misdiagnosis and consequently inappropriate treatment choices [60] (Figure 2).

## 3. MSI in Colorectal Cancer

The Cancer Genome Atlas Network (TCGA) has included the MSI status in the molecular classification of CRC. Particularly, the TCGA classified CRC in different molecular subsets, including the chromosome instability phenotypes (CIN), accounting for 70% of cases, the MSI phenotype, accounting for 20%, and the CpG island methylation phenotype (CIMP), accounting for 10% [61] (Figure 1). Although some studies showed that some MSI CRC cases harbouring also CIMP and CIN overlap with different molecular phenotypes, TCGA classified MSI CRC as a distinct molecular phenotype [62,63]. Approximately 3% to 5% of all CRCs have dMMR associated with LS. This syndrome, known as hereditary non-polyposis colorectal carcinoma (HNPCC), is an autosomal dominant syndrome with germline mutations of the MMR genes [64,65,66]. Beyond the CRCs associated with LS, approximately 70–95% of MSI CRCs are sporadic [67]. The relationship between CRC MSI and some clinical features, such as primary right-sided tumors and older age, has been demonstrated [68]. The MSI status is an early event in the development of CRC; it has never been identified in polyps [69]. On the contrary, MSI has also been observed in the normal colonic mucosa of patients with LS [70,71,72]. Several studies suggest that MSI CRCs at advanced stages show a poor prognosis compared to MSS CRC [73,74]. The clinical studies KEYNOTE-016 and KEYNOTE-164 demonstrated the efficacy of pembrolizumab, with durable responses in patients with advanced MSI-H/dMMR or mCRC treated after chemotherapy [75,76]. In KEYNOTE-177, a phase 3 study, patients with unresectable or metastatic MSI/dMMR CRC were treated with pembrolizumab as the first-line treatment without receiving prior therapy; this showed clinically significant improvements in survival compared to chemotherapy. In 2020, the FDA approved pembrolizumab as the first-line treatment for advanced or metastatic CRC with MSI-H/dMMR [77]. Furthermore, CheckMate 142 demonstrated the efficacy of nivolumab (an anti-PD1 antibody) used alone or in combination with ipilimumab (an anti-CTLA-4 antibody) in mCRC MSI-H/dMMR patients previously treated with chemotherapy. In 2018, the FDA approved nivolumab either alone or in combination with ipilimumab [78,79] (Table 1). The predictive value of MSI in CRC requires the optimization of testing in clinical practice. As already mentioned, IHC is the most used method; however, it showed several discrepancies with PCR and NGS results. Several studies have compared the different assays for MSI detection in CRC series [41,50,80,81,82] (Table 2). The comparative analysis between different methods showed several discrepancies; in particular, some of the dMMR-IHC cases were MSS-PCR (ranging from 1% to 33% of cases) [82,83,84]. In contrast, only 5% of pMMR-IHC cases analyzed in the literature showed MSI by PCR [41,50,80,81,85]. Generally, the PCR results are in agreement with the NGS results, showing a discrepancy of up to 3% [58,83,86,87]. To date, no recommendations have been proposed regarding the gold standard molecular test for MSI analysis. The NGS is the most sensitive method for MSI detection. Moreover, the NGS represents the best molecular approach since it allows simultaneously the detection of both MSI and a wide panel of other mutations. However, PCR can represent a valid molecular test for MSI detection in daily clinical practice since it is economical and widespread in the laboratories compared to NGS. Finally, MSI analysis by NGS could be useful in selected cases showing discordant results between IHC and PCR.

## 4. MSI in Gastric Cancer

The TCGA has included MSI status in the molecular classification of GC. Particularly, the TCGA classified GC in different molecular subsets, including the chromosome instability phenotypes (CIN), accounting for 50% of cases, the genomically stable (GS) tumors, accounting for 19%, the MSI phenotype, accounting for 22%, and the Epstein-Barr virus (EBV)-positive tumors, accounting for 9% (Figure 1) [88]. Although rare cases simultaneously carrying MSI and EBV have been identified, the TCGA considers MSI as a molecular profile distinct from the others [89]. Similarly, the Asian Cancer Research Group (ACRG) classified GC in MSS and MSI [90]. Many studies have found MSI in precancerous lesions and an increase in MSI frequency in GC cases is observed [91,92]. MSI GCs usually occur in older female patients, with an intestinal subtype at an early stage (I/II) and with better differentiation and distal position, and no lymph node involvement [44,91]. Ottini et al. observed that the MLH1 promoter methylation is frequently in sporadic GC, while MLH1 and MSH2 mutations are rare in MSI GCs [93]. MSI occurs both in sporadic GCs and in GCs associated with LS [94,95]. A positive correlation was observed between the presence of Helicobacter pylori (H. pylori) and MSI. The active bacterial infection is present more frequently in GC MSI patients, suggesting that H. pylori may affect the MMR system during the gradual progression of gastric carcinogenesis [52]. Approximately 85% of the MSI GC cases concomitantly showed a KRAS mutation. On the contrary, no association between MSI GC and a mutated BRAF was reported [96]. MSI and human epidermal growth factor receptor 2 (HER2) amplification are reported to be mutually exclusive, suggesting that HER2 amplification could be a negative predictive marker for immunotherapy in GC patients [97]. The MSI status is not used as an exclusive prognostic biomarker, since the prognosis of GC patients is affected by other clinical and pathological features, including the age, stage, grade and chemotherapy treatment [98]. To date, controversial data about the prognostic role of MSI in GC have been reported. Several studies have shown that GCs carrying MSI have a good prognosis [91,99,100]. However, other studies showed MSI not having a prognostic role; in particular, MSI GCs do not have a better survival time than MSI-L/MSS patients. [101,102]. The metastatic MSI GCs have shown long-lasting positive results after receiving treatment with anti-PD-1 antibodies (durvalumab, pembrolizumab and nivolumab), as a single therapy or in combination with anti-CTLA4 antibodies (ipilimumab and tremelimumab) [103]. Pembrolizumab was investigated among patients with advanced MSI gastric or gastroesophageal junction cancer who were enrolled in different studies, such as KEYNOTE-059, KEYNOTE-061 and KEYNOTE-062. The results indicate that pembrolizumab or pembrolizumab plus chemotherapy provided durable antitumor activity compared to chemotherapy alone in patients with an MSI gastric junction or gastroesophageal cancer. Other clinical trials have suggested drug combinations, such as durvalumab and tremelimumab or nivolumab and ipilimumab in phase 2, for the treatment of advanced MSI GC [104,105,106] (Table 3). The comparative analysis between these methods showed some discrepancies, particularly as some dMMR-IHC cases were MSS by PCR (ranging from 1% to 38%) [51,107], while pMMR-IHC cases were MSI by molecular analysis (ranging from 1% to 7,4%) [52,108,109].

The FDA accelerated approval of pembrolizumab in 2017 to treat MSI patients in solid tumors, including GC, who have progressed following prior treatment [27]. Few data have been reported regarding the MSI testing in GC; all the comparison data between IHC and PCR are summarized in Table 4.

## 5. MSI in Gynecologic Cancers

Several studies have shown the presence of MSI from 2 to 20% in ovarian cancer (OC) (Figure 1), particularly in clear cell carcinomas and endometrioid cancers [110,111,112]. In clear cell carcinoma, the expression of CD8, PD-1 and tumor-infiltrating lymphocytes (TIL) was higher in MSI tumors than in those of MSS. In contrast, in endometrioid carcinomas, no correlation between the MSI status and TIL was reported [111,113]. MSI OCs are characterized by advanced stage, poor differentiation, grade and poor prognosis [114]. MSI OCs are frequently associated with LS [115]. However, the correlation between MSI and LS is not always verified since the LS was observed in both MSI OC and MSS OC [116]. MSI detection in OC has not yet been extensively studied and no adequate method is currently defined. The suitability and specificity of the Bethesda panel are still today an open issue in this cancer type [117]. There are few studies that have compared IHC and PCR for detecting MSI in OC (Table 5).

A concordance was shown between the IHC analysis and PCR [113]. Some discordant results have been observed; in particular, dMMR-IHC cases were MSS by PCR in a wide subset accounting from 1% to 80% [118,119,120]. Although MSI is still a poorly understood marker in OC patients, several studies showed its potential predictive role for ICI in this cancer (Table 6). The TCGA has included the MSI status in the molecular classification of endometrial cancer (EC). Particularly, the TCGA classified EC in different molecular subsets, including the no specific molecular profile (NSMP), accounting for 38% of cases, the MSI phenotype, accounting for 30%, the copy number high/mutant TP53 (CNH), accounting for 25%, and the POLE/ultramutated (POLE), accounting for 7% (Figure 1) [121,122,123]. Although some studies showed that some MSI EC cases also harbouring other mutations overlapped with different molecular phenotypes, the TCGA classified MSI EC as a distinct molecular phenotype [124]. LS in the EC was observed with a frequency of 3% [125].

Regarding morphological features, MSI ECs are frequently characterized by the presence of peritumoral lymphocytes and TILs [126]. In approximately 95% of ECs, the MSI is due to the hypermethylation of the MLH1 promoter [127]. MSI ECs showed in about 10–53% of cases the simultaneous presence of other gene alterations, particularly the mutations of the following genes: RPL22 (46%) [128,129,130], PTEN (34%) [128,131], KRAS (35%) [128], ATR (15%) [132], CHK1 (29%) [133], Caspase5 (10%) [131,134] and BAX (27%) [129,134,135]. Furthermore, the PTEN gene mutations, frequently observed in sporadic EC (25–83%), showed a correlation with MSI with a frequency of 60% [136]. Moreover, MSI ECs are associated with a higher neoantigen load and increased TIL, CD3-positive, CD8-positive and PD-L-1 positive compared to MSS/pMMR EC [137]. The role of MSI in ECs is until now unclear; most of the studies have concluded that the MSI status has no prognostic role [138,139,140]. Previous studies have shown that the combination of pembrolizumab and lenvatinib can be used for second-line treatment in patients with EC who are not dMMR/MSI-H, and studies are ongoing for the treatment of patients with advanced EC MSI [141,142]. In the USA, both pembrolizumab and dostarlimab are approved as monotherapy for the treatment of MSI EC patients. In the EU, dostarlimab is the only therapy approved for patients with EC dMMR/MSI who have failed platinum therapy [143] (Table 7). Several studies have compared the different assays for MSI detection in EC series (Table 8).

The comparative analysis between these methods showed some discrepancies; in particular, approximately 1–68% of dMMR-IHC cases were MSS by PCR/NGS [83,86,144,145,146], and about 5% of pMMR-IHC cases were MSI-H by molecular analysis [147,148].

## 6. MSI in Other Malignancies

MSI has been found in several cancer types, including non-small cell lung cancer, melanoma, breast cancer, urothelial cancer, pancreatic ductal adenocarcinoma and brain cancer. The expansion of MSI clinical trials into other cancers may elucidate the prognostic and predictive value of MSI for non-colorectal. The frequency of MSI reported in non-small cell lung cancer (NSCLC) is very heterogeneous, ranging from 0.8% to 40% [149,150,151,152]. The role and clinical implications of MSI are still unclear in NSCLC. Carpagnano et al. suggested that MSI is associated with a poor prognosis in NSCLC [153].

Moreover, recent findings suggested that MSI in NSCLC is associated with an increased response to immunotherapy [154]. Several clinical trials showed the efficacy of pembrolizumab for the treatment of previously treated NSCLC patients carrying MSI [155,156,157] (Table 9).

MSI is extremely rare in breast cancer, showing a frequency of 0.04–3% [158,159]. However, MSI was observed in triple negative breast cancer (TNBC), with a high frequency accounting for between approximately 0.2% and 18,6% [160,161,162]. The role of MSI in BC is still uncertain. Some studies have shown that MSI does not have a significant impact on patient survival [149]. Conversely, other studies have reported a positive prognostic value of MSI in TNBC [160]. A significant correlation was observed between MSI and the negative expression of estrogen and progesterone receptors, indicating a possible relationship between genetic changes in microsatellite regions and hormonal deregulation in the progression of BC [163]. In BC, MSI showed a correlation with the advanced clinical stage, the tumor size and the high grade [164]. The KEYNOTE-028 and KEYNOTE-012 studies demonstrated the efficacy of pembrolizumab in BC previously treated with CDK4/6 inhibitors. [158,165]. The pancreatic ductal adenocarcinoma (PDAC) showed MSI in about 1–2% [166,167]. PDACs carrying MSI are generally associated with LS [168]. LS-associated pancreatic tumors, ranging from 0.5% to 3.9%, are generally associated with medullary histology showing prominent lymphocyte infiltration [167,169]. PDAC is an aggressive type of human cancer; moreover, most patients are resistant to chemotherapy. Thus, immunotherapy could be a possible therapeutic option [170]. Previous studies showed that pMMR PDCA patients treated with conventional chemotherapy have a survival advantage compared to dMMR patients, suggesting that MSI can be considered a potential predictor of treatment sensitivity [171,172,173]. KEYNOTE-016 evaluated the efficacy of pembrolizumab in eight cases of MSI PDAC, showing a complete response in two patients and a partial response in three patients [27,174]. These preliminary results suggested that anti-PD-L1 antibody treatment could be effective in patients with MSI PDAC. MSI was reported in approximately 10–15% of thyroid carcinomas (TC) [175,176,177]; moreover, a correlation between MSI and clinical-pathological features was observed. Particularly, MSI TCs were generally papillary or anaplastic and poorly differentiated histotypes. A recent study showed that MSI TC patients have a good prognosis compared to MSS TC [177]. Although several clinical trials are being evaluated for the efficacy of pembrolizumab, nivolumab and atezolizumab in MSI TC, until now no immunological therapy has been approved for MSI TC [178,179]. The ICI targeting the PD-1/PD-L1 pathway may be a treatment option for MSI TC; however, further studies are needed [177]. The frequency of MSI brain tumors is very low, ranging from 2% to 3% of cases [27,116]. Brain tumors are rarely associated with LS (0.5–3.7% of all cases). No information is available regarding the specific molecular or phenotypic characteristics of LS-related brain tumors [167]. The MSI phenotype was observed in about 25% of glioblastomas (GBM) [180]. MSI GBM showed a low level of PD-L1 expression, suggesting it is not a good predictive marker for ICI targeting the PD1/PD-L1 pathway [181]. MSI has been shown to occur in malignant melanomas, with a frequency of between 2 and 30% [182,183,184]. Kubecek et al. suggested that MSI might be a predictive marker in malignant melanoma [185]. Although the KEYNOTE-016 study showed the efficacy of pembrolizumab in MSI melanomas that have progressed after previous treatment, the treatment for MSI melanoma is not yet used in clinical practice [27].

MSI has been identified in approximately 1–28% of urothelial carcinomas (UC) [186,187,188]. The upper tract urothelial carcinomas (UTUCs) also occur in about 5% of patients with LS [189]. The heterodimer MSH2/MSH6 is frequently lost in the UTUC in about 50–86% of cases [186,190,191]. Several studies reported that UTUCs MSI showed unique morphological features compared to MSS UTUC, including increased intratumoral lymphocytes, the lack of nuclear pleomorphism and the presence of pushing edges [167,190,192]. The KEYNOTE-158 study demonstrated the benefit of pembrolizumab treatment in patients carrying MSI-H/dMMR including UCs [174]. The prostate cancers (PCs) showed MSI status in 1,2–12% of cases. MSI PCs represent a clinically aggressive phenotype [193]. The germline alterations in the MMR genes are less common in localized PC, suggesting an increase in mutations in patients with metastatic spread [194]. In the KEYNOTE-016 study, MSI PCs responded better to ICI [27]. Although MSI can be considered a biomarker capable of responding to immunotherapy in prostate cancer, ICIs are not yet used in clinical practice for the treatment of metastatic PCs [195].

## 7. Conclusions

The MMR system is a key repair mechanism for maintaining sequence fidelity and stability. The inactivation of the MMR system due to the germinal, somatic or epigenetic prevents error correction and, thus, promotes microsatellite instability. MSI has been detected in various cancers with a prognostic and/or predictive role; in particular, it is clinically important for predicting a response to immunotherapy. MSI represents the FDA’s first cancer type-agnostic biomarker approved for selection to the treatment with pembrolizumab of patients with any advanced solid cancer, regardless of the histology. The results reported underline the pivotal role of MSI in the choice of drug treatment in patients, especially in the cases of those who no longer respond to chemotherapy. To date, MSI detection aimed at selecting patients for ICI treatment could be performed through both IHC and PCR/NGS. Although the two tests are equated in clinical practice for the therapeutic choice, several controversial results have been reported. In this view, the optimization of MSI detection in clinical practice is currently needed in order to implement an adequate selection of patients eligible for ICI treatment of dMMR tumors. A feasible diagnostic algorithm for MSI testing could include IHC as prescreening requiring an additional molecular test for cases with an equivocal IHC of one or more MMR proteins, and also to confirm dMMR cases in order to exclude false IHC results.

## Figures and Tables

**Figure 1 ijms-23-08726-f001:**
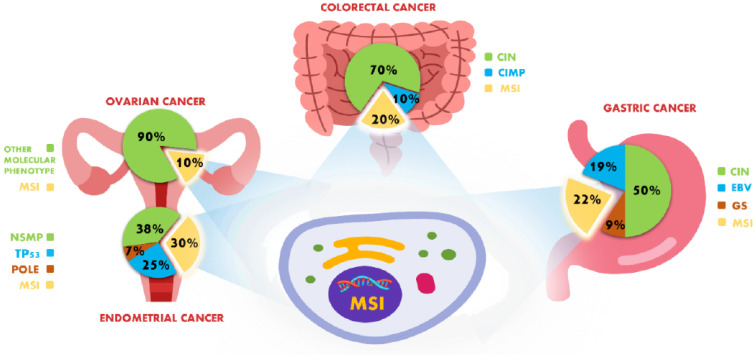
Frequency of microsatellite instability in gastrointestinal and gynecologic cancers. Distribution of MSI according to the molecular classification of colorectal cancer, gastric cancer, ovarian and endometrial cancers. CIN: chromosome instability; CIMP: CpG island methylator phenotype; MSI: microsatellite instability; EBV: Epstein–Barr virus; GS: genomically stable; NSMP: no specific molecular profile; POLE: DNA polymerase epsilon.

**Figure 2 ijms-23-08726-f002:**
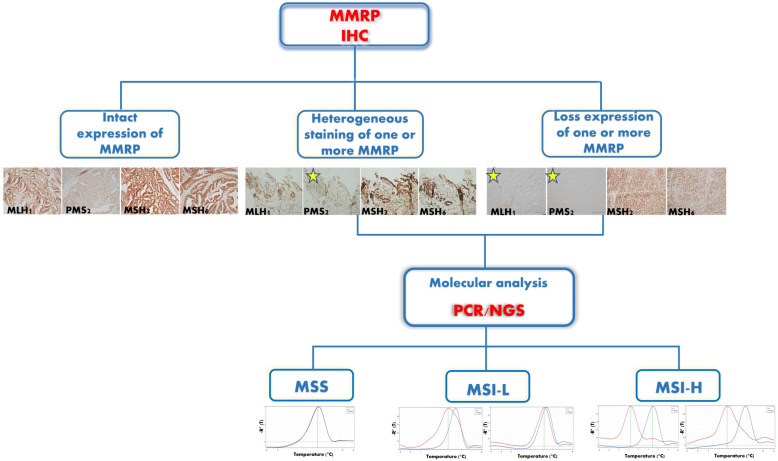
The diagnostic algorithm of MSI status. MMRP IHC is used as a screening test: the intact expression of all four MMRPs is a conclusive result of the MSS status (original magnification 10×); the heterogeneous staining of one or more MMRPs needs an additional molecular assay (original magnification 10×); and the loss of one or more MMRPs needs a molecular assay (original magnification 10×). The molecular analysis performed through PCR and/or NGS leads to a conclusive result in cases with IHC equivocal staining. MMRP: mismatch repair proteins; IHC: immunohistochemistry; PCR: polymerase chain reaction; NGS: new-generation sequencing; MSS: microsatellite stability; MSI-L: microsatellite instability-low; MSI-H: microsatellite instability-high. The stars indicate the PMS2 heterogeneous staining (center) and the loss of MLH1 and PMS2 (right).

**Table 1 ijms-23-08726-t001:** Clinical trials for immune checkpoint inhibitors treatment of MSI CRC.

Clinical Trial	Drug Treatment	Phase	Tumor Type	N. Patients	Status	Conclusion
NCT02460198 KEYNOTE-164	Pembrolizumab	2	Advanced unresectable CRC mCRC Stage IV	124	Completed	Pembrolizumab is effective in MSI-H/dMMR CRC
NCT01876511 KEYNOTE-016	Pembrolizumab	2	MSI-H CRC MSS CRC MSI non-CRC	113	Completed	ORR: disappearance of all target lesions PR: 30% decrease in the diameters of the target lesions
NCT00912743	Olaparib	2	CRC MSI-H	33	Completed	Olaparib activity not demonstrated after failure of standard systemic therapy
NCT03374254 KEYNOTE-651	Pembrolizumab + Binimetinib alone Pembrolizumab + Chemotherapy with or without Binimetinib	1	mCRC MSS/pMMR	220	Active, not recruiting	No results posted
NCT02060188 CheckMate 142	Nivolumab alone vs. Nivolumab + Ipilimumab orNivolumab + BMS-986016 or Nivolumab + Daratumumab Nivolumab + Ipilimumab + Cobimetinib	2	mCRC MSI-H CRC non-MSI-H	385	Active, not recruiting	No results posted
NCT04258111	IBI310 (anti-CTLA-4 antibody) + Sintilimab (anti-PD-1 antibody)	2	CRC MSI-H mCRC MSI-H	4	Active, not recruiting	No results posted
NCT03435107	Durvalumab	2	mCRC MSI-H or POLE	33	Active, not recruiting	No results posted
NCT03350126	Ipilimumab + Nivolumab	2	mCRC MSI/dMMR	57	Active, not recruiting	No results posted
NCT03186326	FOLFOX or FOLFIRI Protocol + Panitumumab + Cetuximab + Bevacizumab + Aflibercept vs. Avelumab	2	mCRC MSI-H	132	Active, not recruiting	No results posted
NCT02563002 KEYNOTE-177	Pembrolizumab vs. standard therapy (mFOLFOX6 and FOLFIRI alone or associated with Bevacizumab + Cetuximab)	3	MSI CRC stage IV	307	Active, not recruiting	Pembrolizumab prolongs PFS or OS
NCT03827044	Avelumab	3	POLE CRC CRC MSI Stage III	402	Active, not recruiting	No results posted
NCT03926338	Toripalimab + Celecoxib vs. Toripalimab alone	1–2	MSI-H/dMMR CRC	34	Recruiting	No results posted
NCT04636008	Sintilimab + Hypofractionated radiotherapy	1–2	Rectal cancer MSI-H/dMMR	20	Recruiting	No results posted
NCT04014530	Ataluren + Pembrolizumab	1–2	mCRC pMMR and dMMR mEC dMMR	47	Recruiting	No results posted
NCT04988191	Toripalimab + Bevacizumab + Irinotecan	1–2	CRC dMMR/MSI-H	44	Recruiting	No results posted
NCT04715633	Camrelizumab + Apatinib	2	Locally advanced dMMR/MSI-H CRC	52	Recruiting	No results posted
NCT03519412	Temozolomide (induction) Pembrolizumab (treatment)	2	mCRC pMMR and dMMR	102	Recruiting	No results posted
NCT05116085	Tislelizumab	2	With early-stage (Stage II-III) MSI-H or dMMR CRC	38	Recruiting	No results posted
NCT04715633	Camrelizumab + Apatinib	2	MSI-H/dMMR CRC	52	Recruiting	No results posted
NCT05118724	Atezolizumab	2	CRC MSI-H/dMMR Stage III	120	Recruiting	No results posted
NCT04695470	Fruquintinib + Sintilimab	2	mCRC TMB and non MSI-H	70	Recruiting	No results posted
NCT03638297	BAT1306 + Aspirin	2	CRC MSI-H/dMMR or TMB	27	Recruiting	No results posted
NCT04730544	Nivolumab + Ipilimumab	2	mCRC dMMR/MSI-H	96	Recruiting	No results posted
NCT04895722	Pembrolizumab alone vs. Pembrolizumab + Quavonlimab vs. Pembrolizumab + Favezelimab vs. Pembrolizumab + Vibostolimab	2	CRC MSI Stage IV	320	Recruiting	No results posted
NCT04301557	PD-1 Antibody + Oxaliplatin + Capecitabine + External beam radiotherapy + Total mesorectal excision	2	Advanced CRC dMMR/MSI-H	25	Recruiting	No results posted
NCT04304209	Oxaliplatin + Capecitabine + Sintilimab + Total mesorectal excision vs. Oxaliplatin + Capecitabine + Sintilimab + Radiotherapy + Total mesorectal excision vs. Oxaliplatin + Capecitabine + Radiotherapy + Total mesorectal excision	2–3	CRC Stage IICRC Stage III dMMR/MSI-H	195	Recruiting	No results posted
NCT02912559	Atezolizumab + Fluorouracil vs. Leucovorin calcium + Oxaliplatin vs. Fluorouracil + Leucovorin calcium + Oxaliplatin	3	CRC dMMR Stage III	700	Recruiting	No results posted
NCT05239741	Pembrolizumab alone vs. 5-fluorouracil alone or 5-fluorouracil + Bevacizumab or 5-fluorouracil + Cetuximab or FOLFIRI + Bevacizumab or FOLFIRI + Cetuximab	3	CRC MSI-H Stage IV dMMR CRC	100	Recruiting	No results posted
NCT02997228	Atezolizumab alone vs. mFOLFOX6 + Bevacizumab vs. mFOLFOX6 + Bevacizumab + Atezolizumab	3	mCRC MSI-H	231	Recruiting	No results posted
NCT04008030	Nivolumab alone vs. Nivolumab + Ipilimumab vs. Active comparator (Oxaliplatin, Leucovorin, Fluorouracil, Irinotecan, Bevacizumab, Cetuximab)	3	mCRC MSI-H	748	Recruiting	No results posted
NCT05236972	Sintilimab alone vs. Oxaliplatin + Capecitabine	3	CRC dMMR/MSI-H Stage III	323	Recruiting	No results posted
NCT05239741	Pembrolizumab alone vs. Active comparator (Oxaliplatin, Leucovorin, 5-fluorouracil, Irinotecan, Bevacizumab, Cetuximab)	3	CRC MSI-H/dMMR Stage IV	100	Recruiting	No results posted
NCT05217446	Pembrolizumab alone vs. Encorafenib + Cetuximab + Pembrolizumab	2	mCRC MSI-H/dMMR	104	Not yet recruiting	No results posted
NCT05217446	Pembrolizumab alone vs. Encorafenib + Cetuximab + Pembrolizumab	2	mCRC MSI-H/dMMR	104	Not yet recruiting	No results posted
NCT05231850	Tislelizumab	2	CRC dMMR/MSI-H Stage II and III	70	No yet recruiting	No results posted
NCT05231850	Tislelizumab	2	CRC dMMR/MSI-H Stage II and Stage III	70	Not yet recruiting	No results posted
NCT04866862	Fruquintinib + Camrelizumab	2	Non MSI-H/dMMR Refractory CRC	32	Not yet recruiting	No results posted
NCT05131919	Pembrolizumab	2	Locally advanced, Irresectable, dMMR not- mCRC	25	Not yet recruiting	No results posted
NCT05215379	Xintilimab injection	2–3	Rectal cancer immunotherapy MSI-L	180	Not yet recruiting	No results posted

CRC: colorectal cancer; mCRC: metastatic CRC; MSS: microsatellite stability; MSI: microsatellite instability; MSI-H: microsatellite instability high; MSI-L: microsatellite instability low; dMMR: deficient mismatch repair; TMB: tumor mutational burden; POLE: DNA polymerase epsilon; ORR: objective response rate; PFS: progression free survival; PR: partial response; CR: complete response; RECIST: response evaluation criteria in solid tumors.

**Table 2 ijms-23-08726-t002:** Comparison between IHC, PCR and NGS in CRC cases.

Cases	IHC Results	PCR Results	NGS Results	References
28	16 dMMR	15 MSI-H	15 MSI	[83]
		1 MSS	1 MSS	
	12 pMMR	12 MSS	12 MSS	
93	135 dMMR	132 MSI-H	NP	[84]
		3 MSS		
	458 pMMR	4 MSI-H		
		10 MSI-L		
		444 MSS		
988	102 dMMR	98 MSI-H	NP	[41]
		4 MSS/MSI-L		
	886 pMMR	4 MSI-H		
		882 MSS/MSI-L		
91	54 dMMR	48 MSI-H	47 MSI	[58]
			1 MSS	
		6 MSS/MSI-L	6 MSS	
	37 pMMR	37 MSS/MSI-L	37 MSS	
73	12 dMMR	8 MSI-H	NP	[50]
		1 MSI-L		
		3 MSS		
	61 pMMR	3 MSI-H		
		11 MSI-L		
		47 MSS		
15	6 dMMR	6 MSI-H	6 MSI	[86]
	1 loMMR	1 MSI-H	1 MSI	
	1 paMMR	1 MSS	1 MSS	
	6 pMMR	6 MSS	1 MSI	
			5 MSS	
	1 NV	1 MSS	1 MSS	
262	28 dMMR	26 MSI-H	NP	[80]
		2 MSI-L		
	234 pMMR	9 MSI-L		
		225 MSS		
296	65 dMMR	63 MSI-H	NP	[82]
		2 MSS		
	7 loMMR	5 MSI-H		
		2 MSI-L		
	224 pMMR	7 MSI-L		
		217 MSS		
98	38 dMMR	33 MSI-H	32 MSI-H	[85]
			1 MSS	
		1 MSI-L	1 MSS	
		4 MSS	4 MSS	
	60 pMMR	1 MSI-H	1 MSS	
		59 MSS	59 MSS	
809	148 dMMR	147 MSI-H	NP	[81]
		1 MSS		
	12 loMMR	11 MSI-H		
		1 MSS		
	649 pMMR	1 MSI-H		
		3 MSI-L		
		645 MSS		
166	75 dMMR	75 MSI-H	NP	[87]
	91 pMMR	90 MSS/MSI-L		
		1 MSI-H		

dMMR: deficient mismatch repair; pMMR: proficient mismatch repair; IHC: immunohistochemistry; PCR: polymerase chain reaction; NGS: next generation sequencing; MSI-H: microsatellite instability-high; MSS: microsatellite stability; MSI-L: microsatellite instability-low; NP: not performed.

**Table 3 ijms-23-08726-t003:** Clinical trials for immune checkpoint inhibitors treatment of MSI GC.

Clinical Trials	Drug Treatment	Phase	Tumor Type	N. Patients	Status	Conclusions
NCT04795661	Pembrolizumab	2	Localized resectable tumor MSI/dMMR or EBV-positive GC	120	Recruiting	No results posted
NCT04817826	Durvalumab + Tremelimumab	2	GC MSI-H	31	Recruiting	No results posted
NCT05177133	Capecitabine + Oxaliplatin + Retifanlimab	2	dMMR Esophagogastric cancer	25	Recruiting	No results posted
NCT04152889	Camrelizumab + S-1 + Docetaxel	2	GC Stage III (PD-L1 +/MSI-H/EBV +/dMMR)	20	Recruiting	No results posted
NCT04006262	Nivolumab + Ipilimumab	2	Localized MSI and/or dMMR Oeso-gastric adenocarcinoma	32	Recruiting	No results posted
NCT03257163	Capecitabine + Pembrolizumab + Radiation therapy	2	dMMR and Epstein–Barr virus Positive GC	40	Recruiting	No results posted

GC: gastric cancer; MSI: microsatellite instability; MSI-H: microsatellite instability-high; dMMR: deficient mismatch repair; EBV: Epstein–Barr virus; PD-L1: programmed death-ligand 1.

**Table 4 ijms-23-08726-t004:** Comparison between IHC, PCR and NGS in GC cases.

Cases	IHC Results	PCR Results	NGS Results	References
56	13 dMRR	8 MSI-H	NP	[107]
		5 MSI-L		
	43 pMMR	43 MSS		
580	61 dMRR	60 MSI-H	NP	[51]
		1 MSS		
	519 pMMR	519 MSS		
60	6 dMRR	6 MSI-H	NP	[52]
	54 pMMR	4 MSI-H		
		50 MSS		
16	9 dMMR	9 MSI-H	NP	[108]
	paMMR	1 MSI-H		
	lo-paMMR	1 MSI-H		
	5 pMMR	5 MSS		
50	4 dMMR	4 MSI-H	NP	[109]
	46 pMMR	44 MSS		
		2 MSI-H		

MMR: deficient mismatch repair; pMMR: proficient mismatch repair; IHC: immunohistochemistry; PCR: polymerase chain reaction; NGS: next generation sequencing; MSI-H: microsatellite instability-high; MSS: microsatellite stability; MSI-L: microsatellite instability-low; NP: not performed.

**Table 5 ijms-23-08726-t005:** Comparison between IHC, PCR and NGS in OC cases.

Cases	IHC Results	PCR Results	NGS Results	References
30	3 dMMR	3 MSI-H	NP	[113]
	27 pMMR	27 MSS		
834	228 dMMR	41 MSI-H	NP	[118]
		187 MSS		
	606 pMMR	83 MSI-H		
		523 MSS		
26	7 dMMR	7 MSI-H	NP	[119]
	19 pMMR	1 MSI-H		
		18 MSS		
42	4 dMMR	4 MSI-H	NP	[120]
	38 pMMR	2 MSI-H		
		3 MSI-L		
		33 MSS		

dMMR: deficient mismatch repair; pMMR: proficient mismatch repair; IHC: immunohistochemistry; PCR: polymerase chain reaction; NGS: next generation sequencing; MSI-H: microsatellite instability-high; MSS: microsatellite stability; MSI-L: microsatellite instability-low; NP: not performed.

**Table 6 ijms-23-08726-t006:** Clinical trials for immune checkpoint inhibitors treatment of MSI OC.

Clinical Trial	Drug Treatment	Phase	Tumor Type	N. Patients	Status	Conclusion
NCT03836352	DPX-Survivac + Cyclophosphamide + Pembrolizumab vs. DPX-Survivac + Pembrolizumab	2	Solid tumors, including OC and MSI-H	184	Active, not recruiting	No results posted

OC: ovarian cancer; MSI-H: microsatellite instability-high.

**Table 7 ijms-23-08726-t007:** Clinical trials for immune checkpoint inhibitors treatment of MSI EC.

Clinical Trial	Drug Treatment	Phase	Tumor Type	N. Patients	Status	Conclusion
NCT04906382	Carboplatin + Paclitaxel + Tislelizumab	1	dMMR EC	20	Recruiting	No results posted
NCT05112601	Ipilimumab + Nivolumab vs. Nivolumab alone	2	dMMR recurrent EC	12	Recruiting	No results posted
NCT02912572	Avelumab alone vs. Avelumab + Talazoparib vs. Avelumab + Axitinib	2	mEC MSI-H	105	Recruiting	No results posted
NCT04774419	Intensity modulated radiation therapy (IMRT) + TSR-042	2	EC dMMR/MSI-H	31	Recruiting	No results posted
NCT05036681	Futibatinib vs. Pembrolizumab	2	MSS mEC	30	Recruiting	No results posted
NCT05173987 Keynote C93	Pembrolizumab alone vs. carboplatin + paclitaxel + docetaxel + cisplatin	3	dMMR EC	350	Recruiting	No results posted
NCT03241745	Nivolumab	2	EC dMMR/MSI-H	35	Active, not recruiting	No results posted
NCT05201547	Dostarlimab alone vs. Carboplatin-Paclitaxel	3	EC dMMR	142	Not yet recruiting	No results posted

EC: endometrial cancer; mCRC: metastatic EC; MSS: microsatellite stability; dMMR: deficient mismatch repair.

**Table 8 ijms-23-08726-t008:** Comparison between IHC, PCR and NGS in EC cases.

Cases	IHC Results	PCR Results	NGS Results	References
108	33 dMMR	27 MSI-H	NP	[144]
		6 MSS		
	75 pMMR	75 MSS		
98	18 dMMR	8 MSI-H	NP	[145]
		10 MSS/MSI-L		
	5 loMMR	2 MSI-H		
		3 MSS/MSI-L		
	75 pMMR	75 MSS		
100	52 dMMR	51 MSI-H	NP	[146]
		1 MSS		
	10 loMMR	6 MSI-H		
		4 MSS		
	9 paMMR	3 MSI-H		
		1 MSI-L		
		5 MSS		
	18 pMMR	18 MSS		
	11 NA	8 MSI-H		
		3 MSS		
89	26 dMMR	23 MSI-H	NP	[147]
		3 MSS		
	63 pMMR	3 MSI-H		
		60 MSS		
99	29 dMMR	NA	16 MSI	[148]
			13 MSS	
	70 pMMR		2 MSI	
			68 MSS	
21	9 dMMR	6 MSI-H	8 MSI	[83]
		2 MSS		
		1 MSI-L	1 MSS	
	3 loMMR	2 MSS	2 MSS	
		1 MSI-H	1 MSI	
	9 pMMR	9 MSS	9 MSS	
15	6 dMMR	5 MSI-H	6 MSI	[86]
		1 NA		
	3 loMMR	2 MSI-H	1 MSI	
			1 MSS	
		1 MSS	1 MSS	
	1 paMMR	1 MSS	1 MSS	
	5 pMMR	5 MSS	5 MSS	

dMMR: deficient mismatch repair; pMMR: proficient mismatch repair; IHC: immunohistochemistry; PCR: polymerase chain reaction; NGS: next generation sequencing; MSI-H: microsatellite instability-high; MSS: microsatellite stability; MSI-L: microsatellite instability-low; NA: not available; NP: not performed.

**Table 9 ijms-23-08726-t009:** Clinical trials for immune checkpoint inhibitors treatment of other MSI/no-MSI-H tumors.

Clinical Trial	Drug Treatment	Phase	Tumor Type	N. Patients	Status	Conclusion
NCT01876511 KEYNOTE-016	Pembrolizumab	2	MSI CRC MSS CRC MSI Non-CRC	113	Completed	ORR: disappearance of all target lesions; PR: 30% decrease in the diameters of the target lesions.
NCT04328740	TP-1454 monotherapy vs. TP-1454 + Ipilimumab + Nivolumab	1	Advanced solid tumor Advanced/metastatic RCC, MSI-H or dMMR mCRC, NSCLC	44	Recruiting	No results posted
NCT02332668 KEYNOTE-051	Pembrolizumab	1–2	Melanoma, Lymphoma, Solid tumor, Classical Hodgkin Lymphoma; MSI-H solid tumor	320	Recruiting	No results posted
NCT04521075	Fecal microbial transplantation by capsules	1–2	Metastatic or inoperable melanoma, MSI-H, dMMR or NSCLC	42	Recruiting	No results posted
NCT03607890	Nivolumab + Relatlimab	2	Advanced dMMR cancers resistant to Prior PD-(L)1 inhibitor	42	Recruiting	No results posted
NCT03667170	Envafolimab	2	dMMR/MSI-H advanced solid tumors	200	Recruiting	No results posted
NCT03236935	L-NMMA + Pembrolizumab	1	NSCLC, Malignant melanoma Head and neck squamous cell carcinoma Classical Hodgkin Lymphoma Urothelial carcinoma Bladder DNA repair-deficiency disorders	12	Active, not recruiting	No results posted
NCT03053466	APL-501	1	Solid tumors MSI-H or dMMR	30	Active, not recruiting	No results posted
NCT04800627	Pembrolizumab + Pevonedistat	1–2	dMMR/MSI-H metastatic or locally advanced unresectable solid tumor	2	Active, not recruiting	No results posted
NCT02983578	Danvatirsen + Durvalumab	2	Advanced and refractory pancreatic, NSCLC and dMMR CRC	53	Active, not recruiting	Not result posted
NCT03241745	Nivolumab	2	MSI/dMMR/Hypermutated uterine cancer	35	Active, not recruiting	No results posted
NCT04326829	QL1604	2	dMMR or MSI-H advanced solid tumors	86	Not yet recruiting	No results posted

CRC: colorectal cancer; mCRC: metastatic CRC; MSS: microsatellite stability; MSI: microsatellite instability; MSI-H: microsatellite instability-high; dMMR: deficient mismatch repair; RCC: renal cell carcinoma; NSCLC: non-small cell lung cancer; ORR: objective response rate; PFS: progression free survival; OS: overall survival; PR: partial response; RECIST: response evaluation criteria in solid tumors.

## Data Availability

Not applicable.

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
