# Peer review of "Microsatellite Instability: From the Implementation of the Detection to a Prognostic and Predictive Role in Cancers"

_ijms, 2022, doi:10.3390/ijms23158726_

Round 1

Reviewer 1 Report

I really appreciate the big efforts by the authors to shed light on a very hot topic of current oncology: the predictive and prognostic role of MSI in cancers, the therapeutic agnostic role of ICIs in MSI status and the available diagnostic tool. However, they should take into account some suggestions: 

- In the abstract, please report for extensive: IHC, PCR, NGS, and please check also in the main text. 

- The English language should be more fluent and readable.  

- In the Introduction section, please clarify better the main mechanisms that can lead to a MSI status, since the sentences about this topic are too rough .

- In the Methods section, line 120, Please re-write this sentence: "MSI-L and MSS tumors have similar molecular and clinical features, thus patients carrying MSI-L are classified as MSS and they are not eligible for immunotherapy", since the eligibility to immunotherapy does not depend only exclusively from the MS status. 

- In the method section, the sentences from line 124 to 126 need to be better explained. 

- Please, Check English language line 131.

- Please, specify the phase 3 trial cited at line 157.

- Please, Check English language line 162.

- For the sentences from line 185 to 188, could the author provide, according to what they state, a kind of diagnostic algorithm in case of test discrepancies? 

- Please re-write the title of paragraph n. 6: MSI in other malignancies (brain cancer are not carcinoma) 

- Please re-formulate the conclusion paragraph stressing better the agnostic role of MSI and the diagnostic algorithm. 

Author Response

- In the abstract, please report for extensive: IHC, PCR, NGS, and please check also in the main text. 

Thank you for the suggestion, we have corrected the abstract and the main text.

- The English language should be more fluent and readable.  

As correctly suggested by the Referee, we have made a massive correction of the English language. All corrections are marked in the text.

- In the Introduction section, please clarify better the main mechanisms that can lead to a MSI status, since the sentences about this topic are too rough.

As correctly observed by the Referee, the mechanism and the biology about MSI status were not well argued. Thus, we improved the Introduction at line 54, as follows:

“MLH1 and MSH2 are the primary partners of their heterodimer, while PMS2 and MSH6 are the respective secondary partners. Consequently, the loss of the primary partners results in the loss of the entire heterodimer, but not the opposite. The alteration of the MMR proteins leads to a biological status known as microsatellite instability (MSI) [8]. The generation of MSI can be caused by the point mutations occurring in MMR genes, the DNA polymerase slippage during the replication process and the insertion/deletion of one or more bases in the microsatellite regions [1].”

- In the Methods section, line 120, Please re-write this sentence: "MSI-L and MSS tumors have similar molecular and clinical features, thus patients carrying MSI-L are classified as MSS and they are not eligible for immunotherapy", since the eligibility to immunotherapy does not depend only exclusively from the MS status. 

As correctly suggested by the referee, we have changed the sentence as follow: “MSI-L and MSS tumors have similar molecular and clinical features, thus patients carrying MSI-L are classified as MSS and they are not sensitive to immunochecpoint inhibitors compared to patients carrying MSI-H”

- In the method section, the sentences from line 124 to 126 need to be better explained. 

As correctly suggested by the referee, we have implemented the main text as follow: “To date, both IHC and PCR are used in the clinical practice to MSI testing in order to define patients who can benefit from ICI treatment. However, several data in the literature showed discordant results between IHC and PCR in the MSI detection. Several studies demonstrated that MMR IHC results frequently have not been confirmed by molecular analysis. For example, some cases dMMR-IHC are results MSS by PCR resulting in clinical implications in the therapeutic choice. Likewise, also cases with pMMMR-IHC were unexpectedly MSI by PCR.  [36,44,45,46,47].”

- Please, Check English language line 131.

Thank you for the suggestion, we have changed the sentences.

- Please, specify the phase 3 trial cited at line 157.

Done

- Please, Check English language line 162.

Done

- For the sentences from line 185 to 188, could the author provide, according to what they state, a kind of diagnostic algorithm in case of test discrepancies? 

We improved the manuscript adding a new Figure related to the MSI diagnostic algorithm, however our flow chart does not discriminate between PCR and NGS since to date no guidelines are provided about the molecular assay to use as gold standard for MSI detection.  As correctly observed by the Referee in case of the discrepancies between PCR and NGS a diagnostic algorithm should be elaborate, however we preferred not include it in the figure since it could be confusing for the reader. We preferred to discuss this critical issue in the text as follows:

“To date, no recommendations have been proposed regarding the gold standard molecular test for MSI analysis. The NGS is the most sensitive method for MSI detection.  Moreover, the NGS represents the best molecular approach since it allows simultaneously the detection both the MSI and a wide panel of other mutations. However, PCR can represent a valid molecular test for MSI detection in daily clinical practice since it is economical and widespread in the laboratories compared to NGS. Finally, the MSI analysis by NGS could be useful in selected cases showing discordant results between IHC and PCR.”

We hope that our review will be satisfactory, we thank the reviewer who allowed us to shed light on such a critical point.

- Please re-write the title of paragraph n. 6: MSI in other malignancies (brain cancer are not carcinoma) 

Thank you for the important suggestion, we have changed the title of the paragraph n.6.

- Please re-formulate the conclusion paragraph stressing better the agnostic role of MSI and the diagnostic algorithm. 

Thank you for the important suggestion, we have improved the conclusion paragraph stressing better both the cancer type-agnostic role of MSI, as follow: “The MSI has been detected in various cancers with a prognostic and/or predictive role, particularly it is clinically important for predicting response to immunotherapy. MSI represents the FDA’s first cancer type-agnostic biomarker approved for selection to the treatment with pembrolizumab of patients with any advanced solid cancer regardless of the histology.”

Furthermore, we have drawn up a summary figure of the diagnostic algorithm and implemented the conclusions paragraph as follows: “To date, the MSI detection aimed at selecting the patients for ICI treatment could be performed through both IHC and PCR/NGS. Although the two tests are equated in clinical practice for the therapeutic choice, several controversial results have been reported. In this view,  the optimization of the MSI detection in clinical practice is currently needed in order to implement an adequate selection of patients eligible for ICI treatment of dMMR tumors. A feasible diagnostic algorithm for MSI testing could include the IHC as prescreening requiring an additional molecular test  for cases with equivocal IHC of one or more MMR proteins and also to confirm dMMR cases in order to exclude false IHC results.”

Reviewer 2 Report

In this manuscript, Amato et al have provided an overview of MSI in different cancers, their detection methods as well as their prognostic and therapeutic indications, especially in the context of checkpoint blockade. Overall, it is a nicely written review that will be of interest and importance to the readers in the field. I recommend following revisions to make this a better manuscript.   

Major:

-          Figure 1: Although figure 1 is cited in the introduction, there should be a bit more elaboration on figure 1 description in the text. For example, it is not clear what the pie charts are trying to depict in figure 1. Are the authors trying to say that, for example, 22% of gastric cancer patients have MSI? In that case, there is no overlap possible for those patients that have other drivers? Like EBV? Overall, the pie charts will need more description and discussion in the text.

-          Line 79 – It would be useful for the readers to have a brief paragraph before line 79 to explain the rationale behind why immunotherapies are likely to be beneficial for patients with MSI.

-          There are too many paragraphs throughout the manuscript (some with 2-3 sentences as well). This distracts the flow of reading. I recommend combining these paragraphs systematically to make fewer paragraphs in each section for the sake of better flow.

Minor:

-          Line 34 – “Provide” will be more appropriate instead of “propose”

-          Line 69 – “In the last time, MSI….” – This can be revised as “In recent years, MSI…”

-          Line 99 – word “spread” seems to be a mistake here? Or please revise the use of the word as it is not clear.

-          Line 168 – This seems to be a typo error -- “stabile”

-          Line 297 – “In the UE..” Is this meant to be EU?

Author Response

Referee 2

In this manuscript, Amato et al have provided an overview of MSI in different cancers, their detection methods as well as their prognostic and therapeutic indications, especially in the context of checkpoint blockade. Overall, it is a nicely written review that will be of interest and importance to the readers in the field. I recommend following revisions to make this a better manuscript.   

Major:

-          Figure 1: Although figure 1 is cited in the introduction, there should be a bit more elaboration on figure 1 description in the text. For example, it is not clear what the pie charts are trying to depict in figure 1. Are the authors trying to say that, for example, 22% of gastric cancer patients have MSI? In that case, there is no overlap possible for those patients that have other drivers? Like EBV? Overall, the pie charts will need more description and discussion in the text.

Thank you, the referee, for the suggestion. We have implemented the text in the different paragraphs for each cancers type, as follows:

“Particularly, the TCGA classified CRC in different molecular subsets, including the chromosome instability phenotypes (CIN) accounting for 70% of cases, the MSI phenotype accounting for 20% and the CpG island methylation phenotype (CIMP) accounting for 10% [56] (Figure 1). Although some studies showed that some MSI CRC cases harbouring also CIMP and CIN overlapping with different molecular phenotypes, however TCGA classified MSI CRC as a distinct molecular phenotype.  (Goel A, et al. Gastroenterology. 2007) (Li LS, et al. Am J Pathol. 2003).”

“The TCGA has included the MSI status in the molecular classification of GC. Particularly, the TCGA classified GC in different molecular subsets, including the chromosome instability phenotypes (CIN) accounting for 50% of cases, the genomically stable (GS) tumors accounting for 19%, the MSI phenotype accounting for 22% and the Epstein-Barr virus (EBV)-positive tumors accounting for 9% [81] (Figure 1). Although rare case carrying simultaneouslyMSI and EBV have been identified, however the TCGA considers MSI as a molecular profile distinct from the others (Yang N, et al. PeerJ. 2021).”

“The TCGA has included the MSI status in the molecular classification of EC. Particularly, the TCGA classified EC in different molecular subsets, including the no specific molecular profile (NSMP) accounting for 38% of cases, the MSI phenotype accounting for 30%, the copy number high/ mutant TP53 (CNH) accounting for 25% and the POLE/ ultramutated (POLE) accounting for 7% [113,114,115] (Figure 1). Although some studies showed that some MSI EC cases harbouring also other mutations overlapping with different molecular phenotypes, however TCGA classified MSI EC as a distinct molecular phenotype. (Cancer Genome Atlas Research Network, Kandoth C, Schultz N, Cherniack AD, Akbani R, Liu Y, Shen H, Robertson AG, Pashtan I, Shen R, Benz CC, Yau C, Laird PW, Ding L, Zhang W, Mills GB, Kucherlapati R, Mardis ER, Levine DA. Integrated genomic characterization of endometrial carcinoma. Nature. 2013) (Bosse T,et al. Am J Surg Pathol. 2018).”

 We hope that our revisions will be satisfactory, we thank the reviewer who allowed us to shed light on such a critical point.

-          Line 79 – It would be useful for the readers to have a brief paragraph before line 79 to explain the rationale behind why immunotherapies are likely to be beneficial for patients with MSI.

 As correctly suggested by the referee, we have explained the rationale behind the sensitivity to ICI of MSI-H tumors, as follow: 

The immunocheckpoint inhibitorshave been shown to be effective in several tumor types carrying MSI-H  compared to chemotherapy. (Le DT, et al 2015) (Jover R, et al. 2009) (Nikanjam M et al 2020). The rationale behind the sensitivity to ICI of MSI-H tumors is probably due to the accumulation of mutations that determines  new mutated peptides leading to the activation of the immune system [Drescher KM et al 2009]. In this context, MSI tumors have a high level of the T-helper 1/ cytotoxic lymphocytes and the immune checkpoint molecules such as CTLA4, PD-1, PD-L1, LAG-3 and IDO, compared to MSS tumours. [Llosa NJ, et al. 2015] [Ma C, et al. 2016].”

-          There are too many paragraphs throughout the manuscript (some with 2-3 sentences as well). This distracts the flow of reading. I recommend combining these paragraphs systematically to make fewer paragraphs in each section for the sake of better flow.

 As correctly observed by the referee, the subdivision into too many subparagraphs can confuse the reader. Thus, we revised all main text and we combined the paragraphs, the revisions are marked in the main text.

Minor:

-          Line 34 – “Provide” will be more appropriate instead of “propose”

Done

-          Line 69 – “In the last time, MSI….” – This can be revised as “In recent years, MSI…”

Done

-          Line 99 – word “spread” seems to be a mistake here? Or please revise the use of the word as it is not clear.

Done

-          Line 168 – This seems to be a typo error -- “stabile”

Done

-          Line 297 – “In the UE” Is this meant to be EU?

Done

Round 2

Reviewer 2 Report

Thanks for addressing the concerns satisfactorily. However, I still find extensive paragraphs that can be merged so that there is no space between the sentences. Otherwise, everything else is revised to be published. 

Author Response

Thank you so much the Referee, for the suggestion. We merged the extensive paragraphs eliminating unnecessary spaces. We hope that our revisions could be satisfactory. 

Thank you so much. 

Best regards 
